The correlation of WDR76 expression with survival outcomes and immune infiltrates in lung adenocarcinoma

Fang Likui
Yu Guocan
Yu Wenfeng
Chen Gang
Ye Bo B1618142@zju.edu.cn
Department of Thoracic Surgery, Affiliated Hangzhou Chest Hospital, Zhejiang University School of Medicine , Hangzhou , China
Mitsouras Katherine
Electronic publication date: 2021 Oct 4
Publication date: 2021
Volume: 9
Electronic Location ID: e12277
Received 2021 May 6; Accepted 2021 Sep 19
Copyright: ©2021 Fang et al.
Copyright year: 2021
Copyright holder: Fang et al.
License: This is an open access article distributed under the terms of the Creative Commons Attribution License, which permits unrestricted use, distribution, reproduction and adaptation in any medium and for any purpose provided that it is properly attributed. For attribution, the original author(s), title, publication source (PeerJ) and either DOI or URL of the article must be cited.
License URL: https://creativecommons.org/licenses/by/4.0/

Keywords: Lung adenocarcinoma, WDR76, Prognosis, Immune infiltrates, TCGA

Funding: The authors received no funding for this work.

==============================
Background

WD repeat domain 76 (WDR76) is a predicted member of the WD40-repeat-containing domain superfamily and possibly involves in various biological processes, but its function in cancers is poorly characterized. This study aimed to evaluate the role of WDR76 in the prognosis and immune infiltrates of lung adenocarcinoma (LUAD).

Methods

WDR76 expressions in LUAD tissues and normal tissues were primarily compared by The Cancer Genome Atlas (TCGA) database, and were validated in cohorts from Gene Expression Omnibus (GEO) database. The associations between WDR76 expression and clinicopathologic characteristics were analyzed. Kaplan–Meier and Cox regression analyses were performed to determine the impact of WDR76 expression on survival outcomes. The protein interaction network of WDR76 was built using STRING website. TIMER and GEPIA databases were used to investigate the correlation between WDR76 expression and immune infiltrates.

Results

WDR76 expression was elevated in LUAD (P < 0.001) and high WDR76 expression was associated with advanced N stage, M stage and pathologic stage. Expectedly, high WDR76 expression significantly correlated with poor survival outcomes and was the independent risk factor for overall survival (OS) (HR 1.468, 95% CI [1.031–2.089], P = 0.033) and disease specific survival (DSS) (HR 1.764, 95% CI [1.095–2.842], P = 0.020). DDB1 and LSH were the important proteins interacting with WDR76. WDR76 expression correlated with CD8+ T cells presence and was also positively associated with levels of inhibitory receptors.

Conclusion

WDR76 expression was involved in the regulation of immune infiltrates and had predictive value for prognosis in LUAD.

Introduction

Lung cancer is one of the most common malignant tumors and the leading cause of cancer-related mortality worldwide (Siegel, Miller & Jemal, 2020; Bray et al., 2018). Non-small cell lung cancer (NSCLC) accounts for more than 80% cases of lung cancer and lung adenocarcinoma (LUAD) is the most frequent histological subtype (Travis et al., 2015). Most cases of LUAD were diagnosed at advanced or metastatic stage and systematic therapy should be advised (Alexander, Kim & Cheng, 2020). Chemotherapy is not always an effective approach, whereas tyrosine kinase inhibitors (TKIs) therapy has shown a good activity in oncogene driven disease (Piper-Vallillo, Sequist & Piotrowska, 2020; Planchard, 2020). In the era of immunotherapy, there are increasing evidences supporting the use of immune checkpoint inhibitors (ICIs) in the systematic therapy (Yang, Yang & Yang, 2020). Despite the encouraging data, there are still some problems such as limited patient response rates and drug resistance. Ferroptosis is a form of regulated cell death that mainly results from iron accumulation and lipid peroxidation (Mou et al., 2019). Ferroptosis could affect the efficacy of chemotherapy, radiotherapy and immunotherapy, and targeting ferroptosis signaling could improve the outcomes from those therapies (Chen et al., 2021; Wu et al., 2020a). Therefore, inducing ferroptosis is becoming a promising strategy in the treatment of NSCLC (Li et al., 2020b; Lou et al., 2021).

WD repeat domain 76 (WDR76) is a predicted member of the WD40-repeat-containing domain superfamily, whose function is poorly characterized in cancers (Yang, Wang & Chen, 2021). A recent research indicated that WDR76 could activate lipid metabolism-associated genes to inhibit ferroptosis in dependent manner of lymphoid-specific helicase (LSH) which acted as an oncogene in lung cancer (Jiang et al., 2017). However, researches focusing on WDR76 are limited, and its role in immune microenvironment and prognosis of LUAD is unclear. This study explored the prognostic significance of WDR76 expression in LUAD through bioinformatics analysis of the clinical characteristics from The Cancer Genome Atlas (TCGA). TIMER and GEPIA databases were used to investigate the associations of WDR76 expression with infiltrated immune cells and their corresponding gene marker sets. Our results demonstrated that high expression of WDR76 was associated with poor prognosis in LUAD, and WDR76 expression positively correlated with not only CD8+ T cell presence but also inhibitory receptors such as PDCD1, LAG3 and TIM-3. Thus, it was plausible that WDR76 could inhibit ferroptosis and reduce T cell cytotoxicity to promote LUAD progression.

Methods

Data source

The data of LUAD patients with the expression of mRNA and matching clinicopathological information were obtained from The Cancer Genome Atlas (TCGA) database (https://portal.gdc.cancer.gov/). The database is publicly open-access and the approval of local ethics committee is not necessary. A total of 535 patients with LUAD and their clinicopathologic characteristics were included in this study. The gene expression profiling data sets (GSE140797, GSE27262 and GSE18842) were obtained from Gene Expression Omnibus (GEO) database (https://www.ncbi.nlm.nih.gov/gds) to verify the differential expression of WDR76 between tumor tissues and normal tissues.

TIMER and GEPIA database analysis

The Tumor Immune Estimation Resource (TIMER) is a public website (http://timer.cistrome.org/) which is a comprehensive resource for systematic analysis of immune infiltrates across 32 cancer types. The correlation of WDR76 expression with the abundance of six types of infiltrating immune cells (CD8+ T cells, CD4+ T cells, macrophages, B cells, dendritic cells and neutrophils) in LUAD was evaluated by TIMER database (Li et al., 2020a).

The Gene Expression Profiling Interactive Analysis (GEPIA) is an online database which provides key interactive and customizable functions including differential expression analysis, correlation analysis, survival analysis, similar gene detection and dimensionality reduction analysis (Tang et al., 2017). The associations of WDR76 expression with multiple markers for immune cells were investigated by GEPIA database.

Protein–protein interaction comprehensive analysis

The Search Tool for the Retrieval of Interacting Genes/Proteins (STRING) is a website (https://string-db.org/) hosting a great collection of integrated and consolidated protein-protein interaction data (Szklarczyk et al., 2021). We obtained the protein–protein interaction (PPI) network information of WDR76 by the online tool STRING. The confidence score > 0.7 was considered significant.

Statistical analysis

The patients were divided into two groups according to the median expression of WDR76. The measurement data was statistically analyzed with t test or Mann–Whitney U test according to the data distribution. As for the numeration data, chi-square test or corrected chi-square test or the Fisher’s exact test was used, depending on the actual situation. Multiple testing correction was performed by Bonferroni method when testing WDR76 and immune cell markers in GEPIA. Kaplan–Meier curve and log-rank test were performed to analyze the survival impact of WDR76 expression. Survival outcomes included overall survival (OS), disease specific survival (DSS) and progress free survival (PFS). Cox regression analysis was performed to verify the effect of WDR76 expression on survival outcomes. All the above analysis was conducted by R statistical software (version 3.6.3) and SPSS software (version 24.0). Statistical significance was set at P value < 0.05 (All P values presented were 2-sided).

Results

Baseline characteristics

A total of 535 LUAD samples and 59 normal tissue samples from the TCGA database were incorporated in the study. The baseline characteristics of patients including age, gender, smoking history and pathological stage were summarized in Table 1.

Table 1 Baseline characteristics of the LUAD patients.

Characteristic		N	(%)	
Age	≤65	255	49.4	
	>65	261	50.6	
Gender	Female	286	53.5	
	Male	249	46.5	
Smoking history	No	75	14.4	
	Yes	446	85.6	
T stage	T1	175	32.9	
	T2	289	54.3	
	T3	49	9.2	
	T4	19	3.6	
N stage	N0	348	67.1	
	N1	95	18.3	
	N2	74	14.3	
	N3	2	0.4	
M stage	M0	361	93.5	
	M1	25	6.5	
Pathologic stage	Stage I	294	55.8	
	Stage II	123	23.3	
	Stage III	84	15.9	
	Stage IV	26	4.9	
Notes.

The definitions of T, N, M are primary tumor, regional lymph nodes and distant metastasis, respectively.

High WDR76 expression in LUAD

The gene expression level of WDR76 was significantly higher in tumor samples than that in normal tissue samples in TCGA database (P < 0.001, Fig. 1A). In the analysis of associations between WDR76 expression and clinicopathologic characteristics in LUAD (Figs. 1B–1H), the results showed that no significant association of WDR76 expression with age (P = 0.156) and smoking history (P = 0.277). Higher expression of WDR76 was observed in male (P = 0.012), advanced N stage (P = 0.017) and M stage (0.007). Comparing with T1 stage, WDR76 expression significantly increased in T2 stage (P = 0.008), but the difference was not observed in T3 and T4 stage. In addition, WDR76 expression also increased in advanced pathologic stage (Stage III and IV) in comparison to early stage (Stage I and II) (P = 0.003).

Figure 1 WDR76 expression status in LUAD.

(A) WDR76 expression was higher in LUAD tissues than in normal tissues. (B) There was no statistical difference between WDR76 expression and age. (C) WDR76 expression was higher in male than in female. (D) There was no statistical difference between WDR76 expression and smoking history. (E) WDR76 expression was higher in T2 stage than in T1 stage, but the difference was not observed in T3 and T4 stage. (F–H) High WDR76 expression was associated with advanced N stage, M stage and pathologic stage. ns, P ≥ 0.05; ∗P < 0.05; ∗∗P < 0.01; ∗∗∗P < 0.001.

In order to verify the differential expression, we further compared WDR76 expression between LUAD and normal tissues in data sets GSE140797 (7 pairs tumor and normal tissues), GSE27262 (25 pairs tumor and normal tissues) and GSE18842 (44 pairs tumor and normal tissues) from GEO database (Figs. 2A–2C). The results showed increased expression of WDR76 in LUAD compared to normal tissues (P = 0.009, 0.020 and <0.001, respectively).

Figure 2 Validation of higher WDR76 expression in LUAD than that in normal tissues in (A) GSE140797, (B) GSE27262 and (C) GSE18842 datasets. ∗P < 0.05; ∗∗P < 0.01; ∗∗∗P < 0.001.

High WDR76 expression showing poor survival outcomes

Kaplan–Meier survival analysis showed that the patients with high expression of WDR76 correlated with inferior overall survival (OS) (P = 0.004), disease specific survival (DSS) (P = 0.002) and progress free survival (PFS) (P = 0.003), as shown in Figs. 3A–3C. In the univariate Cox model, both high WDR76 expression and advanced pathologic stage were a negative predictor for OS, DSS and PFS. Gender, age and smoking history were not associated with survival outcomes. Multivariate Cox regression analysis further proved that WDR76 expression was the independent risk factor for OS (hazard ratio (HR) 1.468, 95% confidence interval (CI) [1.031–2.089], P = 0.033) and DSS (HR 1.764, 95% CI [1.095–2.842], P = 0.020) (Fig. 4).

Figure 3 Kaplan–Meier survival curves for (A) overall survival (OS), (B) disease specific survival (DSS) and (C) progress free survival (PFS) of the LUAD patients with high and low WDR76 expression level.

Figure 4 Cox regression analysis of WDR76 expression and clinicopathologic characteristics with survival outcomes in LUAD.

(A) Univariate Cox regression analysis in overall survival (OS). (B) Multivariate Cox regression analysis in OS. (C) Univariate Cox regression analysis in disease specific survival (DSS). (D) Multivariate Cox regression analysis in DSS. (E) Univariate Cox regression analysis in progress free survival (PFS). (F) Multivariate Cox regression analysis in PFS.

Constructing protein interaction networks

STRING tool was used to analyze the PPI network of WDR76 protein to determine their interactions. The top 10 proteins and corresponding gene symbols, annotations and scores were listed in Fig. 5. WDR76 protein most significantly interacted with DNA damage-binding protein 1 (DDB1), followed by lymphoid-specific helicase (HELLS, LSH), thymocyte nuclear protein 1(THYN1) and DNA repair protein complementing XP-C cells (XPC).

Figure 5 The protein interaction network of WDR76.

Association between WDR76 expression and immune infiltrates

We explored the association between WDR76 expression and six kinds of infiltrating immune cells (CD8+ T cells, macrophages, neutrophils, B cells, CD4+ T cells and dendritic cells) in LUAD from the TIMER database (Fig. 6). The results showed significant positive associations of WDR76 expression with CD8+ T cells (R = 0.269, P = 1.34e−09), macrophages (R = 0.221, P = 7.39e−07) and neutrophils (R = 0.353, P = 6.50e−16), and negative associations with B cells (R = −0.22, P = 8.15e−07) and CD4+ T cells (R = −0.096, P = 3.34e−02), and no association with tumor purity and dendritic cells.

Figure 6 Correlation of WDR76 expression with immune infiltration in LUAD from the TIMER database.

To further explore the possible role of WDR76 expression in immune infiltrates, GEPIA database was used to assess relationships between WDR76 and immune marker sets of various immune cells. The results showed that WDR76 expression was statistically associated with the levels of some immune sets (Table 2). It should be noted that inhibitory receptors such as PDCD1, LAG3 and Tim-3 were significantly and positively associated with WDR76 expression.

Table 2 Correlation analysis between WDR76 and markers of immune cells in GEPIA.

Cell type	Gene marker	Normal	Tumor	
		R	Adj. P	R	Adj. P	
B cell	CD19	0.14	0.9	−0.062	0.51	
KRT20	0.24	0.2	0.041	1	
CD38	0.23	0.25	0.064	0.48	
CD8+ T cell	CD8A	0.21	0.24	0.2	1.7e−05	
CD8B	0.11	0.84	0.21	4.8e−06	
Tfh	BCL6	0.25	0.16	−0.0041	1	
ICOS	−0.076	1	0.12	0.033	
CXCR5	−0.091	1	−0.12	0.033	
Th1	TBX21	0.42	0.006	0.098	0.22	
STAT4	0.39	0.016	0.096	0.24	
IL12RB2	0.21	0.7	0.4	2.3e−19	
IL27RA	0.26	0.32	0.012	1	
STAT1	0.45	0.0024	0.48	1.5e−28	
IFNG	0.22	0.62	0.3	6e−11	
TNF	0.15	1	0.02	1	
Th2	GATA3	0.36	0.021	0.16	0.002	
CCR3	0.24	0.27	−0.042	1	
STAT6	0.38	0.012	−0.11	0.072	
STAT5A	0.32	0.048	0.077	0.36	
Th9	TGFBR2	0.38	0.009	−0.098	0.096	
IRF4	0.0081	1	−0.013	1	
SPI1	−0.24	0.2	−0.00019	1	
Th17	STAT3	0.37	0.018	0.13	0.015	
IL21R	0.34	0.034	0.13	0.014	
IL23R	0.046	1	0.048	1	
IL17A	−0.0046	1	0.13	0.014	
Th22	CCR10	0.16	0.46	0.055	0.46	
AHR	0.28	0.06	0.088	0.11	
Treg	FOXP3	0.21	0.36	0.11	0.06	
IL2RA	0.15	0.72	0.26	1.3e−08	
CCR8	0.22	0.27	0.15	0.0039	
Macrophage	CD68	−0.21	0.22	0.14	0.0038	
CD11b	0.13	0.68	0.00082	1	
M1	NOS2	0.39	0.0069	0.075	0.3	
IRF5	0.029	1	0.12	0.022	
PTGS2	0.048	1	0.14	0.0069	
M2	CD163	−0.21	0.48	0.18	2.4e−04	
ARG1	0.15	1	0.023	1	
MRC1	−0.12	1	−0.066	0.6	
MS4A4A	−0.27	0.17	0.068	0.52	
TAM	CCL2	0.019	1	0.13	0.012	
CD80	0.19	0.64	0.1	0.092	
CD86	−0.12	1	0.15	0.0048	
CCR5	0.33	0.039	0.14	0.0068	
Monocyte	CD14	0.1	1	0.13	0.011	
FCGR3B	0.3	0.07	0.24	3.3e−07	
CD115	0.18	0.54	0.08	0.23	
Neutrophil	CD66b	−0.0049	1	−0.24	1.5e−07	
FUT4	0.43	0.0023	0.22	4.5e−06	
CD11b	0.13	1	0.00082	1	
Natural killer cell	XCL1	0.034	1	0.19	0.00009	
CD7	0.15	0.72	0.17	0.00036	
KIR3DL1	0.19	0.42	0.086	0.17	
Dendritic cell	CD1C	0.062	1	−0.33	1.9e−13	
CD141	0.25	0.18	−0.1	0.075	
CD11c	−0.031	1	0.046	0.93	
T cell exhaustion	PDCD1	0.28	0.12	0.18	0.0003	
CTLA4	0.25	0.22	0.11	0.076	
LAG3	0.4	0.0068	0.26	2.5e−08	
TIM-3	−0.21	0.44	0.14	0.008	
Notes.

Adj adjusted

Tfh follicular helper T cell

Th T helper cell

Treg regulatory T cell

TAM tumor-associated macrophage

Discussion

WDR76 is a nuclear WD40 protein, which possibly involves in a variety of distinct biological processes including DNA damage repair, cell cycle progression, apoptosis, gene expression regulation and protein quality control (Yang, Wang & Chen, 2021; Dayebgadoh et al., 2019). Studies aimed at unveiling novel DNA methylation readers have shown that WDR76 specifically binds 5-hydroxymethylcytosine (5hmC) and it acts as a specific reader of 5hmC, hence suggesting involvement of WDR76 in epigenetic transcriptional regulation (Spruijt et al., 2013). Despite the possible involvement of WDR76 in multiple biological processes, its exact role in cancers remains to be elucidated.

Therefore, we conducted the study to explore the effect of WDR76 expression on the progression and prognosis of LUAD on the basis of various databases including TCGA, GEO, TIMER and GEPIA. The results showed that WDR76 expression significantly increased in LUAD in comparison to normal tissues. Similarly, high WDR76 expression was observed in advanced N stage, M stage and pathologic stage, while the effect was not obvious in T stage, which might be caused by the relatively small samples of T3 and T4 stage (49 and 19, respectively). Interestingly, WDR76 expression was higher in male than in female without different expression in smoking history, whose mechanism needed to be further studied. In addition, Kaplan–Meier survival analysis revealed that low expression of WDR76 in LUAD had favorable OS, DSS and PFS compared with high expression. WDR76 expression was further proven to be an independent predictive factor of prognosis by Cox regression analysis. These links to LUAD have been supported by the evidence of other databases. The OncoLnc database (http://www.oncolnc.org/) shows correlation of aberrant WDR76 expression with shorter survival in LUAD. Furthermore, the Human Protein Atlas lists WDR76 as an unfavorable prognostic marker in various cancers including lung cancer (Uhlen et al., 2017).

Given the vital role of WDR76 expression in LUAD, determining the protein interaction network of WDR76 was of high importance. DDB1 and LSH were identified as the important proteins which interacted with WDR76 based on the analysis from STING website. DDB1 is required for DNA repair and has been reported to involve in conferring tolerance to DNA replication stress to promote LUAD development (Liu et al., 2017). LSH, a member of the ATP-dependent helicase in sucrose nonfermenting 2 (SNF2), plays an important role in inhibiting ferroptosis and promoting lung tumorigenesis (Wu et al., 2020b). A recent study showed that LSH could contribute the recruitment of WDR76 to the metabolic gene promoters such as stearoyl-CoA desaturase 1(SCD1) and fatty acid desaturases 2 (FADS2) to upregulate their expressions, which could affect the intracellular levels of iron and lipid reactive oxygen species (ROS) to block ferroptosis (Jiang et al., 2017). WDR76 was also identified as a molecular inhibitor of the Cullin-4 RING ubiquitin ligase (CRL4) system for stability control of LSH and as a crucial regulator in epigenetic regulation of ferroptosis (Huang et al., 2020).

In recent years, immunotherapy with immune checkpoint inhibitors (ICIs) has revolutionized the clinical management of patients with NSCLC, but only a minority of patients experience durable response (Berghmans et al., 2020). A recent report indicated that immunotherapy-activated CD8+ T cells could enhance specific lipid peroxidation in tumor cells to induce ferroptosis, and increased ferroptosis could contribute to the antitumor efficacy of immunotherapy (Wang et al., 2019). However, the interaction between ferroptosis and immune system is complicated, and does not just lead to a completely positive or negative effect (Wu et al., 2020a). In this study, we tried to reveal whether WDR76 played an important role in immune infiltrates by performing bioinformatics analysis of public data. In the TIMER database, WDR76 expression showed negative correlation with B cells and CD4+ T cells. Infiltrating B cells and CD4+ T cells could reside within the tertiary lymphoid structures which correlate with better prognosis in patients with NSCLC (Germain et al., 2014). Our findings also suggested that WDR76 expression was associated with CD8+ T cells presence and its corresponding markers in LUAD. Despite the vital role of CD8+ T cells in antitumor immunity, an efficient antitumor immune response requires the cooperation of both CD8+ and CD4+ T cells (Altorki et al., 2019). Moreover, markers of T cell exhaustion such as PDCD1, LAG3 and Tim-3 were also strongly associated with WDR76 expression. Increased expression of these inhibitory receptors correlated with T-cell dysfunction and disease progression in NSCLC (Thommen et al., 2015). Although the effect of PD-1 blockade correlated with the amount of PD-1 expression, elevated LAG-3 expression was associated with insensitivity to PD-1 axis blockade (Datar et al., 2019). In addition, WDR76 expression was also positively associated with markers of regulatory T cell (Treg) which could directly suppress the antitumor function of CD8+ T cells (Ganesan et al., 2013). T cell receptor (TCR) -based tracking in lung cancer revealed that many of the proliferative exhausted CD8+ T cells shared the same TCR sequences (paired α and β chains) with cells with exhausted phenotypes (Ren et al., 2021). The induction of the exhaustion states of these tumor-infiltrating CD8+ T cells could be attributable to the inhibitory ligands and suppressive immune cells (Ren et al., 2021).

There were notable limitations in our study. First, the information on relative protein levels or downstream pathways involving WDR76 were unable to be provided because of only the RNA sequencing data obtained from the TCGA database. Second, pathologic stage and M stage were not independent prognostic factors, which might be caused by data heterogeneity. Finally, in TIMER and GEPIA databases, the correlation analysis between CD8+ T cell infiltration and tumor progression was unavailable. It was not clear from the analysis if the CD8+ T cell infiltration was the effect of tumor progression or WDR76 expression, although some studies revealed no significant association between CD8+ T cell density and pathologic stage in NSCLC (Donnem et al., 2015; Schulze et al., 2020). Our results need to be validated by further researches.

Conclusion

Our preliminary findings showed that WDR76 was upregulated in LUAD, and high WDR76 expression was associated with clinical progression and considered as an independent risk factor for poor prognosis. WDR76 was also associated with the regulation of immune infiltrates in LUAD. Although WDR76 expression correlated with CD8+ T cells presence, it was positively associated with levels of inhibitory receptors. This study sheds light on WDR76 as a potential prognostic marker for LUAD.

Additional Information and Declarations

Competing Interests

Author Contributions

Data Availability

The authors declare there are no competing interests.

Likui Fang conceived and designed the experiments, performed the experiments, analyzed the data, prepared figures and/or tables, authored or reviewed drafts of the paper, and approved the final draft.

Guocan Yu performed the experiments, prepared figures and/or tables, authored or reviewed drafts of the paper, and approved the final draft.

Wenfeng Yu and Gang Chen analyzed the data, prepared figures and/or tables, authored or reviewed drafts of the paper, and approved the final draft.

Bo Ye conceived and designed the experiments, prepared figures and/or tables, authored or reviewed drafts of the paper, and approved the final draft.

The following information was supplied regarding data availability:

Data is available at NCBI GEO: GSE10799 and GSE27262.

The expression profile and clinical data are available at the TCGA database (https://portal.gdc.cancer.gov/) (level 3 HTseq-FPKM format from the LUAD (Lung Adenocarcinoma) project).

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
