# Peer review of "The correlation of WDR76 expression with survival outcomes and immune infiltrates in lung adenocarcinoma"

_PeerJ, doi:10.7717/peerj.12277_

## Round 0.1 · original submission · Major Revisions

Your manuscript was considered interesting and valuable by the reviewers. However, they raised some major concerns that need to be addressed. One issue is the need for novel functional validation to support your conclusions, rather than drawing conclusions solely relying on correlations which do not prove causality. Additionally, one of the reviewers raised concerns about the sample size of your controls and the fact that it is significantly smaller than that of the tumor samples. There was also an issue regarding your interpretation of the analysis shown in figure 6. Another concern was regarding your conclusion that WDR76 expression can identify response to immunotherapy, and how your data does not support WDR76 increasing infiltration by CD8+ T cells.

Please, submit a detailed rebuttal which shows where and how you have taken all comments and suggestions into consideration. If you do not agree with some of the reviewers’ comments or suggestions, please explain why. Your rebuttal will be critical in making a final decision on your manuscript. Please, note also that your revised version may enter a new round of review by the same or by different reviewers. Therefore, I cannot guarantee that your manuscript will eventually be accepted.

·

Basic reporting

In the submitted manuscript, Fang et al. analyzed the expression of WD repeat domain 76 (WDR76) with tissues from Lung Adenocarcinoma patients to understand the correlation of WDR76 with cancer progression. At the same time, the authors analyzed the WDR76 levels to various databases to understand its correlation with various protein expressions related to cancer progression and immune infiltrates to suggest the potential functions of WDR76 in the regulation of immune infiltrates and cancer progression.
Overall, the manuscript has been with explicit language, providing a clear introduction of the significance of the study.

Experimental design

The experiments designed are well defined, addressing relevant questions. In addition, the authors provided detailed descriptions of the methodologies performed for the rigorous investigation.

Validity of the findings

The findings are interesting and highlight a current gap of knowledge of how WDR76 expression influences cancer progression.
A minor comment on the conclusion of the TIMER analysis is that while stating that WDR76 expression induces infiltration of CD8 T cells is counter-logical. Understandably the data show a positive correlation between WDR76 expression and CD8 T cells’ presence in the tissue. Infiltration of CD8 T cells indicates a positive immune response, which the data do not show. A suggestion for the authors will be to conclude the data as a positive correlation between WDR76 expression and CD8 T cells’ presence rather than using a strong word as ‘infiltration’.

Additional comments

However, certain points warrant further attention from the authors for a better reading of the manuscript.
Figure 1 is broken into two parts and discussed in two sections in the results, creating a discontinuity. Consider changing the text or figure sequencing for a better and linear flow of the paper for the readers.
Several times, the authors have used abbreviations without stating the full form. For terms like OS, DSS, and PFS, the authors used the full forms in the methods section, but please use the full forms first in the result section for better readability.
For other jargons, such as P, T, M stages, define the term and provide the context for addressing those stages in the analysis.
In figure 4, Uni or multivariate Cox analysis, further detailed analysis of the result is warranted. There are various parameters shown in the figure that was never discussed during the analysis.
In Figure 5, please consider listing the proteins based on the scores rather than alphabetic order for a better sense of the figure.
In the TIMER database analysis, the authors analyzed the WDR76 expression with the levels of various immune infiltrates. Simultaneous analysis of the cancer progression with the immune infiltrates is strongly advised to differentiate between the effects of tumor progression versus WDR76 expression. For example, it is not clear from the analysis if the CD8 T cell infiltration is the effect of tumor progression or WDR76 expression. This will also help understanding if the WDR76 expression of the CD8 T cell infiltrates or only the exhaustion of the CD8 T cells, as shown with GEPIA analysis.
Similarly, for CD4 T cell infiltration in TIMER analysis, consider analyzing Th1 cells and Treg cells separately. That would provide a better opportunity for parallel analysis of gene expression as done with GEPIA database.

Reviewer 2 ·

Basic reporting

In this manuscript, Fang et. al, investigate the role of WDR76 as a potential therapeutic target for LUAD. Overall, the article is well written, clear and unambiguous in most parts, though I would recommend proofreading by prefessional copywriter for occasional grammatical mistakes and inaccurate sentence construction. Proper background, literature references and context have been provided where relevant. I have some comments which I think the authors should work on before the manuscript is ready for publication.

Experimental design

All the bioinformatics and statistical analyses have been performed logically and accurately. The question is well defined and the analyses they perform accurately test their hypotheses. The method of analyses have been described with sufficent details in the method section.

Validity of the findings

I have major concern about how the authors interpreted their analyses of the TIMER and GEPIA database (Figure 6). They conclude that WDR76 expression showed positive correlation with some infiltrating immune cells (CD8+ T cells, macrophages, neutrophils), but negative with some (B cells and CD4+ T cells) but offer no explanation regarding why this is happening. Also the authors claim that WDR76 expression can be helpful in identifying patient response to immunotherapy. While this might be true, the contradictory role of WDR76 in increasing CD8+ T cell infiltration and at the same time increasing the markers of T cell exhaustion is confusing and not backing up the authors' claim.
This issue needs to be addressed more clearly in the discussion section (Lines 216-226), More emphasis should given on observations that back their claim (T reg cells, exhaustion markers, etc) and mention the contradictory effect of CD8 infiltration at the end.

Some minor comments:
1. Line 146 - Please write the full form of OS, HR, DSS, PFS and CI here.

2. Figure 1 and figure 2, better to show the individual data points for the graphs.

3. Do the authors have any insight on why in fig 1E, the only significant difference is observed between T1 and T2 stage and not between T1 and the subsequent advanced stages? This should be addressed either in the results or in the discussion.

4. In figure 4, it will be helpful if the authors have a subheading above the tables indicating that A and B show analyses for OS, C and D for PFS and E and F for DSS.

Reviewer 3 ·

Basic reporting

no comment

Experimental design

no comment

Validity of the findings

no comment

Additional comments

Fang et al investigated the role of WDR76 regulation in LUAD and its microenvironment and suggested that the expression of WDR76 was upregulated in lung tumors may have predictive value for LUAD prognosis. The manuscript requires major revision. One of the major concerns is that the authors made bold conclusions, such as “WDR76 was involved in the regulation of immune infiltrates and a potential therapeutic target”, based on preliminary correlation analyses without further functional validation.

1. The authors compared the expression of WDR76 between tumor (n=535) and normal samples (n=59) from the TCGA cohort. The sample size of matched normal samples is much smaller than the tumors, and the sample size of the two validation data sets (GSE10799 and GSE27262) was not mentioned in the text. It is important to exclude tumors without matched normals and perform a paired tumor-normal comparison based on a combined cohort (TCGA, GSE10799 and GSE27262) to confirm the increased expression of WDR76 in the tumors.
2. It is unclear how the WDR76 expression high and the expression low group was defined. It is particularly important for the survival analyses, as the two groups didn’t separate the overall survival very well, even though the p value was significant.
3. Multiple testing correction is needed when testing WDR76 and immune cell markers in GEPIA.
4. WDR76 may have predictive value for prognosis, but the authors didn’t test WDR76 inhibition in cancer cells, and therefore it is hard to conclude that WDR76 can be a potential therapeutic target as stated in the abstract.
5. The authors need to be careful drawing bold conclusions without sufficient experimental evidence. As an example, data supporting the conclusion “WDR76 was also confirmed to be involved in the regulation of immune infiltrates in LUAD” is based on some correlation analyses without experiments testing the causality. WDR76 might be a biomarker but its association with immune infiltration does not necessarily mean that WDR76 directly regulates tumor immunogenicity.
6. The writing of the manuscript needs improvement. P values and effect sizes should be cleared stated following every association test.

---

## Round 0.2 · accepted · Accept

Thank you for thoroughly addressing the reviewers' comments. As a result, your manuscript is much improved.

·

Basic reporting

The authors have responded sufficiently to the issues raised in the review.

Experimental design

Experimental designs were rigorous and well defined.

Validity of the findings

The authors' response to the review and corresponding changes in the manuscript has strengthened the paper.

Additional comments

With the additional revisions, the manuscript is ready to be accepted for publication.

Reviewer 2 ·

Basic reporting

No comment

Experimental design

No comment

Validity of the findings

No comment

Additional comments

The authors have addressed all my concerns and the manuscript is ready for publication.